# "Like a doctor, like a brother": Achieving competence amongst lay health workers delivering community-based rehabilitation for people with schizophrenia in Ethiopia

Laura Asher[1]*, Rahel Birhane[2], Solomon Teferra[2], Barkot Milkias[2], Benyam Worku[2], Alehegn Habtamu[2], Brandon A. Kohrt[3], Charlotte Hanlon[2,4,5]

1 Division of Epidemiology and Public Health, School of Medicine, University of Nottingham, Nottingham, United Kingdom, 2 Department of Psychiatry, College of Health Sciences, School of Medicine, Addis Ababa University, Addis Ababa, Ethiopia, 3 Division of Global Mental Health, Department of Psychiatry, George Washington University, Washington DC, United States of America, 4 Centre for Innovative Drug Development and Therapeutic Trials for Africa (CDT-Africa), College of Health Sciences, Addis Ababa University, Addis Ababa, Ethiopia, 5 Psychology and Neuroscience, Health Service and Population Research Department, King's College London, Institute of Psychiatry, Centre for Global Mental Health, London, United Kingdom

* laura.asher@nottingham.ac.uk

**Data Availability Statement:** All relevant data are within the Figshare repository at https://doi.org/10.6084/m9.figshare.13615418.

## Abstract

### Background

There are gaps in our understanding of how non-specialists, such as lay health workers, can achieve core competencies to deliver psychosocial interventions in low- and middle-income countries.

### Methods

We conducted a 12-month mixed-methods study alongside the Rehabilitation Intervention for people with Schizophrenia in Ethiopia (RISE) pilot study. We rated a total of 30 role-plays and 55 clinical encounters of ten community-based rehabilitation (CBR) lay workers using an Ethiopian adaptation of the ENhancing Assessment of Common Therapeutic factors (ENACT) structured observational rating scale. To explore factors influencing competence, six focus group discussions and four in-depth interviews were conducted with 11 CBR workers and two supervisors at three time-points. We conducted a thematic analysis and triangulated the qualitative and quantitative data.

### Results

There were improvements in CBR worker competence throughout the training and 12-month pilot study. Therapeutic alliance competencies (e.g., empathy) saw the earliest improvements. Competencies in personal factors (e.g., substance use) and external factors (e.g., assessing social networks) were initially rated lower, but scores improved during the pilot. Problem-solving and giving advice competencies saw the least improvements overall. Multimodal training, including role-plays, field work and group discussions, contributed to

**Funding:** This work was supported by the Wellcome Trust [grant number 100142/Z/ 12/Z] Fellowship in International Health awarded to LA. https://wellcome.org/ The RISE project is part of the PRogramme for Improving Mental health care (PRIME), which is funded by the UK Department for International Development (DfID) for the benefit of LMIC (HRPC10). https://www.gov.uk/government/organisations/department-for-international-development/about/research BAK was supported by the U.S. National Institute of Health (K01MH104310). https://www.nih.gov/ The funders had no role in study design, data collection and analysis, decision to publish, or preparation of the manuscript. The content is solely the responsibility of the authors and does not necessarily represent the official views of any of the funders.

**Competing interests:** The authors have declared that no competing interests exist.

early development of competence. Initial stigma towards CBR participants was reduced through contact. Over time CBR workers occupied dual roles of expert and close friend for the people with schizophrenia in the programme. Competence was sustained through peer supervision, which also supported wellbeing. More intensive specialist supervision was needed.

## Conclusion

It is possible to equip lay health workers with the core competencies to deliver a psychosocial intervention for people with schizophrenia in a low-income setting. A prolonged period of work experience is needed to develop advanced skills such as problem-solving. A structured intervention with clear protocols, combined with peer supervision to support wellbeing, is recommended for good quality intervention delivery. Repeated ENACT assessments can feasibly and successfully be used to identify areas needing improvement and to guide ongoing training and supervision.

## Introduction

Non-specialist health workers are increasingly regarded as the cornerstone of equitable access to mental health care in low and middle-income countries (LMIC) [1]. Non-specialists may be health professionals without formal mental health training, including doctors or nurses, or non-professionals such as lay health workers or traditional healers [2]. There is increasing evidence of the effectiveness [2], acceptability and feasibility [3,4] of mental health interventions delivered by non-specialists in LMIC. It has been demonstrated that lay people with no prior mental health experience can, after a few weeks training and adequate ongoing supervision, deliver effective psychological interventions for common mental disorders [5,6] and community care for people with schizophrenia [7]. Yet, there are important gaps in our understanding of the quality of these interventions and the factors that contribute to the competence of lay health workers delivering them.

Therapist competence is *"the extent to which a therapist has the knowledge and skill required to deliver a treatment to the standard needed for it to achieve its expected effects."* [8]. There are several ways that clinical skills can be assessed, progressing from "knows" and "knows how", assessed by multiple choice questions and clinical vignettes, to "shows" and "does", assessed, for example, by standardized role plays and observed treatment sessions, respectively [9,10]. In LMIC there has been a tendency to evaluate only the "knows" and "knows how" aspects, for example the knowledge, attitudes [11] or self-perceived competence [4,12] of non-specialists following training [8,13]. More recently instruments have been developed to assess the competence of lay health workers, such as through the observation of standardised role-plays in Nepal, Pakistan, Liberia, and Uganda ("shows") [13–16], and therapy quality, for example through the observation of counselling sessions for harmful alcohol use and depression in India ("does") [17].

At present there is limited understanding of the best way to train, support, and supervise lay health workers delivering components of mental healthcare in LMIC to maximise competence, and consequently patient outcomes, particularly where a shortage of specialist personnel may also limit supervisory capabilities [7,18,19]. Recent initiatives have highlighted the need to develop tools and guidelines for quantifying the competence level achievable by lay health

workers, in order to achieve and sustain competence in the delivery of mental health interventions [19]. Competency-based approaches lead to tailored supervision and targeted supplementary training to address skill gaps. However, there is limited experience of this process in practice.

Lay health workers need competence to effectively deliver community-based rehabilitation (CBR) services for people with schizophrenia. CBR is a promising intervention to address the complex health, social and economic needs of people with schizophrenia and is recommended as an adjunct to facility-based treatment, including prescription of anti-psychotic medication, in LMIC [20,21]. The Rehabilitation Intervention for people with Schizophrenia in Ethiopia (RISE) project was a pilot study and cluster-randomised controlled trial of CBR [22–24]. The RISE CBR intervention comprises home-based psychosocial support along with community awareness-raising and mobilisation. In RISE, the intervention was delivered by lay persons who were trained in problem solving and basic counselling skills to become CBR workers. The pilot study demonstrated that CBR for schizophrenia is largely acceptable and feasible [23].

To better understand the process of achieving competence among lay health workers in the RISE study, we addressed the following four aims:

1. To adapt a tool to assess the competence of lay health workers delivering CBR in a rural African setting;

2. To describe the level of competence achievable over 12 months for lay health workers delivering the RISE CBR intervention;

3. To explore the factors shaping CBR worker competence; and

4. To appraise the use of competence measures to inform training and supervision.

## Methods

### Study design

We conducted a 12-month mixed methods study using quantitative data to assess CBR worker competence and qualitative data to explore factors shaping competence.

### Setting

This study was part of the RISE project, which aimed to develop, pilot and evaluate the effectiveness of CBR for people with schizophrenia in Ethiopia. RISE was nested in the Programme for Improving Mental healthcarE (PRIME), a research consortium aiming to generate evidence on the integration of mental health into primary care [25]. PRIME, and the embedded RISE study, were set in Sodo district in the Southern Nations, Nationalities and Peoples' Region of Ethiopia. Sodo is a largely rural district where over 90% inhabitants work as subsistence farmers or small scale traders and 48% of the population are not literate [26]. In Ethiopia, the PRIME district mental health plan included interventions at community, facility and organisational levels [27]. At the facility level, health officers and nurses at the eight primary care health centres were trained to diagnose and treat mental disorders including schizophrenia using the World Health Organisation (WHO) Mental Health Gap Action Programme (mhGAP) intervention guide, modified for the setting. Delivery of mental health in primary care is especially important because the nearest specialist mental health care is available at a psychiatric nurse-led outpatient clinic 30–50 km away in the neighbouring district. RISE was one of the initiatives at the community level. CBR was selected specifically as a bridge between

the community and facility-based services for people with schizophrenia. The development and design of the CBR programme is described elsewhere [22]. In the current mixed-methods study, we focus on evaluating competence of lay health workers delivering CBR [22,23].

## Selection and training of community-based rehabilitation workers

Lay persons selected for CBR training had to fulfil the following criteria: (1) completed tenth grade education (secondary school), (2) resident in Sodo district, and (3) expressing an interest in community work. All workers were expected to speak Amharic. Previous experience of community-based work and high attainment in school examinations were desirable. Degree level nurses were excluded. There were 220 applications in response to local adverts, which were ranked and 50 selected for examination using the criteria. The selection examination included multiple-choice questions on appropriate ways to support people with severe mental illness. Twenty applicants proceeded to the interview on the basis of ranked scores, from which 12 successful applicants were selected. One CBR worker dropped out after two weeks training. Another CBR worker exclusively supported a man with intellectual disability during the RISE pilot; it was anticipated his work would require different competencies, hence his scores are not reported in this paper. Of the ten CBR workers presented here, the mean age was 23 years (range 20–37) and there was an equal gender split. Half had some experience in community health work, though none had experience in mental healthcare provision (Table 1).

CBR workers received five weeks training, which was equally split between the classroom and community-based practice (S1 Appendix). Interactive classroom teaching, including group work, viewing and discussion about 'good' and 'bad' communication skills videos, role-plays and quizzes, was delivered in Amharic by coordinators from an Ethiopian CBR project supporting children with disabilities and psychiatrists. Community-based practice included shadowing trained CBR workers at this project, observing psychiatric nurse-led clinics and

**Table 1. CBR worker characteristics and mean (SE) ENACT-E scores[a] across all items during the RISE CBR worker training and RISE pilot study, ranked by score in pilot months 5–11.**

| CBR worker ID | | | Mean item score across all items (SE) | | |
|---|---|---|---|---|---|
| | Years education | Previous community work | Training[b] | Pilot months 0–3[c] | Pilot months 5–11[d] |
| 4 | 11 | Yes | 2.55 (0.13) | 2.87 (0.04) | 2.97 (0.03) |
| 3 | 11 | Yes | 2.55 (0.08) | 2.74 (0.26) | 2.93 (0.15) |
| 2 | 10 | Yes | 2.15 (0.21) | 2.89 (0.08) | 2.92 (0.03) |
| 7 | 12 | None | 2.32 (0.17) | 2.63 (0.09) | 2.91 (0.06) |
| 10 | 11 | Yes | 2.22 (0.14) | 2.73 (0.14) | 2.89 (0.06) |
| 1 | 13 | Yes | 2.30 (0.11) | 2.95 (0.05) | 2.87 (0.09) |
| 9 | 10 | None | 2.27 (0.19) | 2.55 (0.07) | 2.84 (0.06) |
| 8 | 12 | None | 2.36 (0.09) | - | 2.81 (0.08) |
| 5 | 10 | None | 2.23 (0.16) | 2.58 (0.04) | 2.81 (0.04) |
| 6 | 12 | None | 2.06 (0.14) | 2.71 (0.11) | 2.80 (0.14) |
| Mean score across all CBR workers | | | 2.30 (0.05) | 2.75 (0.04) | 2.87 (0.02) |

[a]The ENACT-E (Enhancing Assessment of Common Therapeutic Factors- Ethiopia) structured observational rating scale assesses competence in delivering psychosocial interventions. 21 items cover the domains of therapeutic alliance, personal factors, external factors & other factors. Each item is rated 1 (needs improvement), 2 (done partially) or 3 (done well).

[b]Includes ENACT assessments of role plays roles conducted during CBR worker training at weeks 1, 3 and 5.

[c]Includes ENACT assessments of CBR sessions with persons with schizophrenia participating in the RISE pilot study at months 0, 1 and 3.

[d]Includes ENACT assessments assessments of CBR sessions with persons with schizophrenia participating in the RISE pilot study at months 5, 7 and 11.

making home visits to persons with schizophrenia being followed up for a separate research project. The training was closely tied to the RISE manual, which was prepared following in-depth intervention development work [22] and translated into Amharic (S2 Appendix). The manual covered (a) information about schizophrenia, anti-psychotic medications, disability and human rights, (b) basic counselling and problem solving skills, needs assessment, goal setting, basic risk assessment and steps to deliver the four core and 11 optional modules and (c) protocols and flow charts for difficult situations such as suicidal ideation. The training covered approaches to support CBR worker wellbeing, including how to keep appropriate boundaries with participants, and how safety would be assessed and maintained. Competence assessment included written tests (multiple choice questions and vignettes) at week 3 and week 5 of the training and standardized role-play assessments at weeks 1, 3 and 5 (see competency role-plays described below). CBR workers were to be excluded if they scored <40% on the written test, but all were above the threshold so proceeded to the pilot.

## RISE pilot study participants and intervention

One male and one female supervisor were recruited based on minimum criteria of a diploma level qualification and experience in community work. The pilot study was overseen by an intervention coordinator (RB). One supervisor supported five CBR workers each; and each CBR worker delivered the 12-month CBR programme to one person with schizophrenia and their family. Five men and five women with schizophrenia, aged between 19 and 60 years, were included. All female participants were unable to read and write, whilst male participants had between five and eight years of school education. The mean duration of illness was 10 years (range one to 30 years). At recruitment into the pilot, half of participants were treatment naïve and only one participant was currently taking anti-psychotic medication. Most participants went on to use anti-psychotic medication for varying periods during the 12-month pilot study [23]. The intervention comprised home visits (initially weekly then reducing to monthly), community awareness raising and mobilisation and family support groups [22]. CBR workers attended fortnightly to monthly one-on-one meetings with their supervisors and monthly peer group supervision sessions to share challenges and strategies. In addition, supervisors conducted unannounced observed home visits every two months. Top-up training sessions were conducted every three months, focusing on areas identified through supervision and quality assessments. All home visits were made with caregivers present and CBR workers were required to carry a mobile phone at all times. CBR supervisors made regular risk assessments of the home visit environment. Additional safety procedures, such as only joint visits with supervisors being permitted, were implemented according to the assessment outcome.

## Competence measure

CBR worker competence was measured using the ENhancing Assessment of Common Therapeutic factors (ENACT) rating scale. The ENACT was designed to evaluate competence in non-specialists delivering psychological therapies across cultural settings, focusing on the common therapeutic factors perceived to be essential for successful outcomes [13]. The tool has demonstrated good psychometric properties. Around half of the 18 items integrate culturally-specific issues, such as explanatory models [13]. Items may be grouped into four broad domains: therapeutic alliance (e.g., verbal and non-verbal communication), patient factors (e.g., functioning assessment), external factors (e.g., assessment of life events), and other elements (e.g., promotion of hope, problem solving). Items are rated as 1 (needs improvement), 2 (done partially well), 3 (done well) or not applicable.

Following a procedure outlined for contextualization and adaptation of the ENACT [13], we modified the tool for use in Ethiopia (ENACT-Ethiopia, henceforth ENACT-E) to assess the competence of CBR workers and primary care workers delivering mental health interventions in this setting. Adaptation comprised: (i) an initial workshop including Ethiopian psychiatrists, psychiatric and primary care nurses and CBR supervisors to review an initial Amharic translation of the ENACT-E. The utility, cultural appropriateness and translation of each item and their rating definitions was discussed along with the need for any additional items to capture competencies specific to CBR or primary care. The ENACT-E was revised accordingly. (ii) At a follow-up workshop, participants independently rated three videoed role-plays of CBR sessions using the ENACT-E. Discrepancies between scores were discussed and amendments made to items as required. And (iii) further modifications were made following piloting during the RISE training.

There were three types of alterations. First, cultural adaptations, for example the indication for 'needing improvement' for the item 'non-verbal communication' was adjusted from 'no eye contact' to 'does not make appropriate eye contact', to reflect the fact that in Ethiopian culture lack of eye contact may sometimes be respectful and appropriate. Second, the removal of technical terms inappropriate to CBR worker training (for example 'agency and pathways thinking') or simplification of terminology ('explains that it is common' instead of 'normalize'). Third, the creation of new items to cover the scope of CBR and primary care competencies, including assessment of medication adherence and linkages to social networks. Physical health and substance use assessments were split into separate items to reflect their importance and to ensure they were differentiated. The final ENACT-E had 21 items (S3 Appendix).

## Procedures for assessing competence

The ENACT can be used to assess a standardized role play comparable to an objective structured clinical evaluation (OSCE) in medical training. A person acting as a client is trained to perform in a standardized manner consistent with real-life clients, but with specific prompts to elicit the competencies. Raters are trained to objectively rate the interaction by scoring on the ENACT tool. The ENACT can also be used to rate sessions with real-life clients (instead of trained actors). When the ENACT is used with standardized clients, this assesses 'competence' whereas rating of real-life sessions with actual clients provides a rating of one aspect of intervention quality.

In the current study, CBR worker competence was assessed using the ENACT-E in these two modalities: observed role plays (competence) and routine CBR sessions (intervention quality). The ENACT-E was first used to assess CBR worker competence during standardized role-plays at the end of weeks one, three and five of the training. Role plays lasted 15 minutes and aimed to represent portions of typical CBR sessions. The assessors were psychiatrists and CBR supervisors who had attended the ENACT workshops and a research assistant was trained to act as a standardized client. Assessors selected 'not-applicable' for items not covered in the role plays. Role-plays were doubled rated using video recordings. Discrepancies in scores of double rated sessions were discussed with raters and minor alterations made to the translation or content of items to increase clarity. The ENACT-E was subsequently used to assess CBR workers during routine CBR sessions at the homes of participants at the initial visit and months one, three, five, seven and 11 of the pilot for all CBR workers. The raters were the CBR supervisors, who used the results to give immediate feedback to CBR workers.

## Analysis of competence data

A descriptive analysis was completed using Stata 15.0. For each time point, we generated means and standard errors for each item (including all CBR workers) and mean item scores for each CBR worker (across all items). CBR workers who were given a 'not applicable' rating for a particular item were excluded from item mean scores for that time point. Items classified as 'not applicable' were likewise excluded from CBR worker mean scores for that time point. Double-rated competence assessments were averaged. Summary means were generated for each time point and also by grouping all role play assessments during the training period, and all CBR session assessments in the RISE pilot months 0–3 and months 5–11. To examine patterns in competence relating to type of competency, item means were grouped into therapeutic alliance, personal factors, external factors and other factors (e.g., promotion of hope) for graphical representation. There was no *a priori* threshold ENACT score used as a cut-off for competence vs. incompetence.

## Factors influencing competence

CBR worker attendance at supervision and refresher training sessions was recorded. Six focus group discussions (FGDs) were conducted with CBR workers across three time points; two immediately post-training, two at pilot midline (six months) and two at pilot endline (12 months). There were five CBR workers in each FGD, and each CBR worker participated at each time point. The two CBR supervisors participated in in-depth interviews (IDIs) at pilot midline and endline. FGDs lasted between 96 minutes and 240 minutes. Topic guides covered training modality, content and assessment; experiences of delivering CBR; and perceived competence. In addition, as part of the broader RISE pilot evaluation, CBR participants (seven people with schizophrenia and eight caregivers) participated in IDIs at pilot month 2 and pilot month 12. One interview component explored participants' views of their CBR worker's knowledge and skills, and the quality of their relationship. The pilot FGDs and IDIs also covered broader aspects of acceptability, feasibility and impact of CBR, which are reported elsewhere [23,28]. A research assistant experienced in qualitative research conducted the IDIs and FGDs, all of which were audio recorded; audio recordings were transcribed in Amharic and translated into English. A thematic analysis was conducted using NVivo for Mac to manage the data [29]. An inductive approach was employed to identify themes. In general themes were identified by examining the surface (semantic) meanings of the data. After coding two transcripts an initial coding scheme was created. Iterative adjustments were made to the coding scheme as it was applied to subsequent manuscripts. Codes were then collated into themes by seeking repeated patterns of meaning across the dataset [30]. Deviant cases were identified and incorporated into the framework. A selection of the full transcripts was reread to confirm that the final thematic framework adequately reflected the data collected. We triangulated the quantitative and qualitative findings, seeking to identify areas of convergence and divergence between data sources.

## Ethical considerations

Ethical approval was obtained from the London School of Hygiene and Tropical Medicine (LSHTM) Research Ethics Committee and the Addis Ababa University College of Health Sciences Institutional Review Board. Written informed consent was sought from each participant.

## Results

### Competence ratings

A total of 85 ENACT-E assessments were completed: 30 role-play competence assessments and 55 CBR session quality assessments. Five CBR session assessments were omitted due to logistical difficulties accessing CBR participants' homes, giving a range of between six and nine assessments per CBR worker. There was clear improvement in CBR worker competence over the training period (Table 2). The mean ENACT score across all items was 1.97 (standard error, SE 0.07) at week 1, indicating that initially, on average, CBR workers were assessed as needing improvement (rating 1), but close to completing items partially well (rating 2). However, by week 3 of training the mean ENACT score across all items rose to 2.50 (SE 0.04). CBR worker competence continued to improve throughout the pilot with mean ENACT scores of 2.77 (SE 0.04) at month 5 and 2.95 (SE 0.03) at month 11. This indicates that by the end of the pilot CBR workers were assessed as completing items close to 'well', on average. There were some differences in rating level by type of competency. In general, therapeutic alliance

**Table 2. Mean ENACT-E scores by item for all CBR workers grouped into RISE CBR worker training, RISE pilot months 0–3 and RISE pilot months 5–11.**

| ENACT item | | Mean ENACT score (SE) | | |
| --- | --- | --- | --- | --- |
| | | Training[a] | Pilot months 0–3[b] | Pilot months 5–11[c] |
| **Therapeutic alliance** | | | | |
| 1 | **Non-verbal communication** | 2.85 (0.05) | 2.96 (0.04) | 3 (0) |
| 2 | **Verbal communication** | 2.51 (0.10) | 2.62 (0.11) | 2.87 (0.06) |
| 3 | **Building trust** | 2.75 (0.06) | 2.96 (0.04) | 3 (0) |
| 4 | **Normalization of feelings** | 2.14 (0.09) | 2.58 (0.10) | 2.77 (0.09) |
| 5 | **Empathy** | 2.36 (0.09) | 2.77 (0.08) | 3.0 (0) |
| 17 | **Confidentiality** | 2.10 (0.19) | 2.94 (0.06) | 3 (0) |
| **Personal factors** | | | | |
| 6 | **Impact on functioning** | 1.94 (0.13) | 2.54 (0.13) | 2.93 (0.07) |
| 8 | **Coping mechanisms** | 1.87 (0.12) | 2.56 (0.13) | 2.50 (0.16) |
| 10 | **Substance use** | 1.33 (0.24) | 2.71 (0.11) | 2.76 (0.12) |
| 18 | **Risk assessment** | 1.55 (0.14) | 2.80 (0.14) | 2.86 (0.10) |
| 19 | **Medication adherence** | 2.53 (0.10) | 2.88 (0.07) | 3 (0) |
| 21 | **Physical health** | 1.67 (0.33) | 2.56 (0.16) | 2.83 (0.10) |
| **External factors** | | | | |
| 7 | **Causal model** | 2.19 (0.19) | 2.83 (0.17) | 2.89 (0.06) |
| 9 | **Life events** | 1.73 (0.13) | 2.36 (0.22) | 3 (0) |
| 11 | **Family involvement** | 2.54 (0.10) | 2.90 (0.07) | 3 (0) |
| 20 | **Role of social networks** | 2.35 (0.17) | 2.86 (0.10) | 2.97 (0.03) |
| **Other domains** | | | | |
| 12 | **Goal setting** | 2.11 (0.11) | 2.67 (0.13) | 2.93 (0.05) |
| 13 | **Promotion of hope** | 2.42 (0.10) | 2.78 (0.09) | 2.90 (0.06) |
| 14 | **Psychoeducation** | 2.69 (0.07) | 2.92 (0.05) | 2.97 (0.03) |
| 15 | **Problem solving** | 2.12 (0.12) | 2.0 (0.24) | 2.45 (0.14) |
| 16 | **Giving advice & eliciting feedback** | 2.12 (0.10) | 2.74 (0.10) | 2.77 (0.09) |
| | **Mean score across all items** | 2.30 (0.05) | 2.75 (0.04) | 2.87 (0.02) |

[a]Includes ENACT assessments of role plays roles conducted during CBR worker training at weeks 1, 3 and 5.

[b]Includes ENACT assessments of CBR sessions with persons with schizophrenia participating in the RISE pilot study at months 0, 1 and 3.

[c]Includes ENACT assessments assessments of CBR sessions with persons with schizophrenia participating in the RISE pilot study at months 5, 7 and 11.

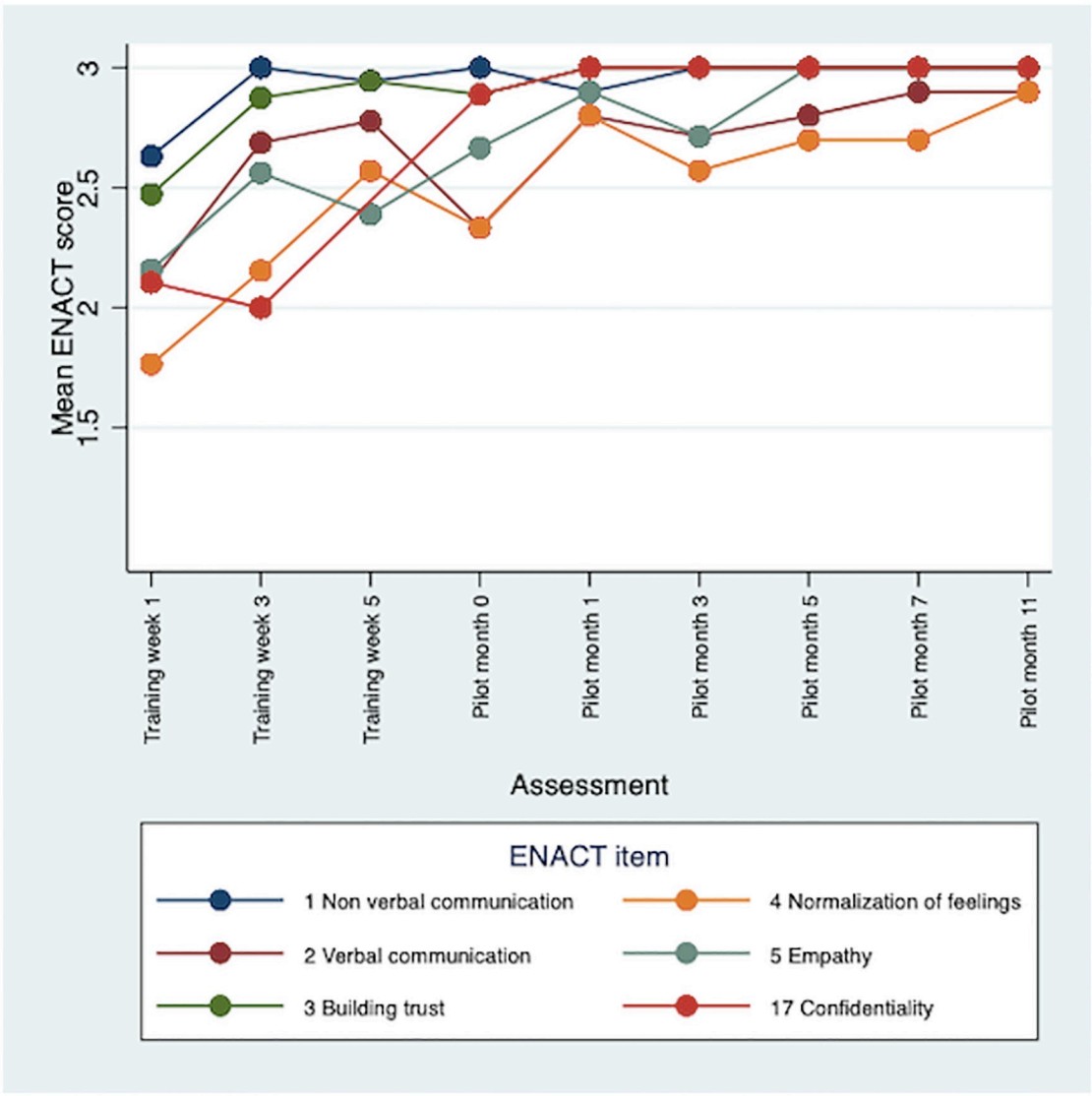

**Fig 1. Mean ENACT scores over time for therapeutic alliance items.**

competencies saw the earliest and greatest improvements (see Fig 1). Items relating to personal factors (e.g., substance use and physical health) and external factors (e.g., assessing social networks) were initially rated lower, but scores improved during the pilot (see Figs 2 and 3). Competencies in the other domains saw the least improvements overall, e.g., promotion of hope showed limited improvement; psychoeducation, however, demonstrated major improvement, with a mean of 3.0 by month 11 (see Fig 4). In response to low ratings by month five (Table 2), top up training was given on skills relating to problem solving, goal setting and giving advice/eliciting feedback; and guidance was given on how to explore life events, coping mechanisms and impact on functioning. In addition, home visit forms were amended to include compulsory sections on substance use and physical health. By months 5–11 of the pilot study, there were improvements in assessing functioning (2.93), life events (3.0), and goal setting (2.93). Scores relating to problem solving, assessing coping mechanisms and giving advice had

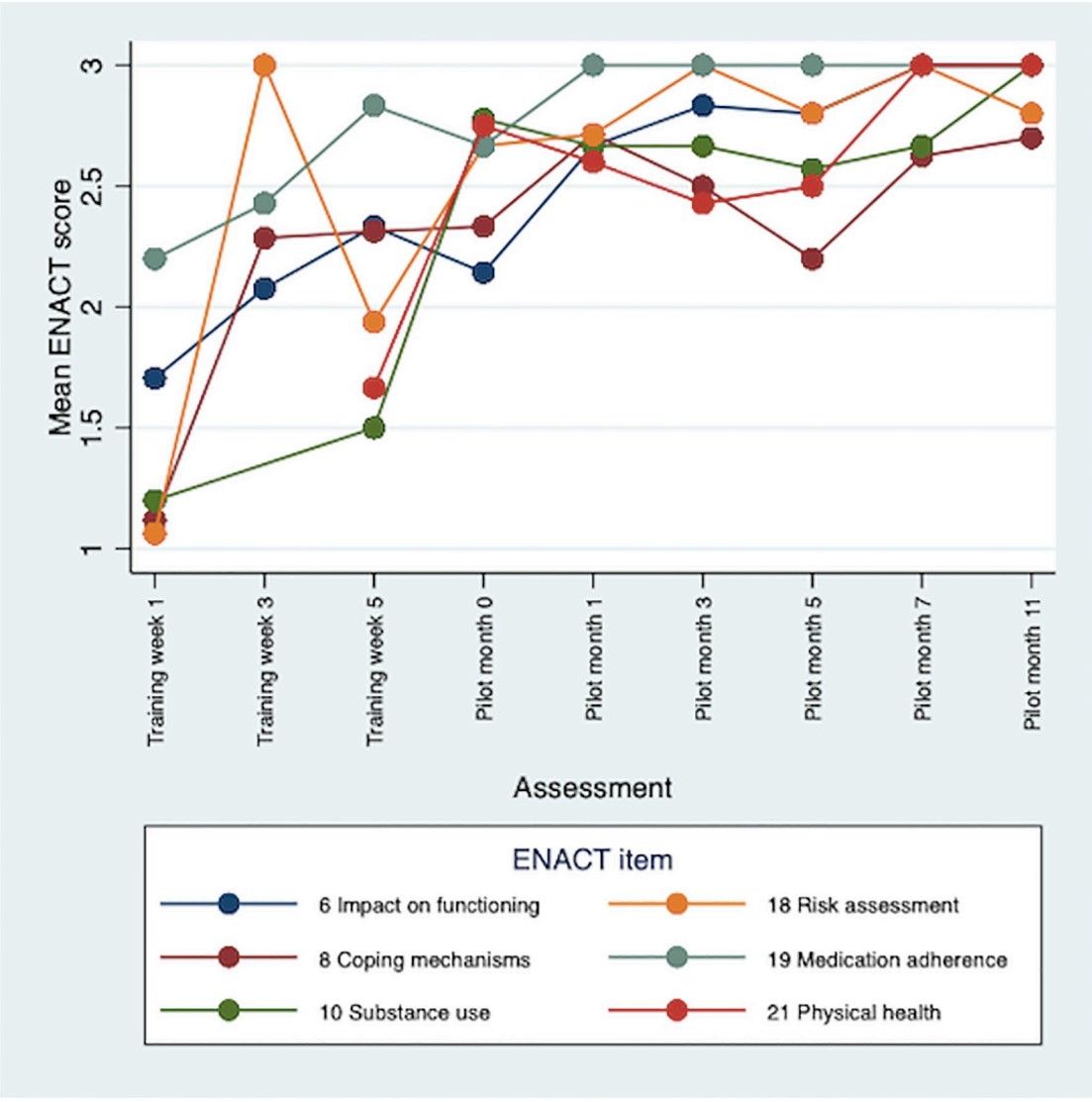

**Fig 2. Mean ENACT scores over time for personal factors items.**

improved somewhat by months 5–11, but these remained some of the lowest rated items (2.45, 2.50 and 2.77, respectively).

There were some differences in competence between CBR workers, with overall consistency in rankings of CBR workers across the training and pilot. Across the role-play training assessments, mean ENACT scores ranged from 2.06 (CBR worker 6) to 2.55 (CBR workers 3 and 4) (Table 1). The gap had narrowed during months 5–11 of the pilot, with mean scores ranging from 2.80 (CBR worker 6) to 2.97 (CBR worker 4). In general, those with previous relevant experience tended to score higher at all time points (Table 1).

### Factors influencing competence

Three main themes were identified: (i) developing competence through training, (ii) challenges in translating theory to practice and (iii) sustaining competence through supervision and wellbeing.

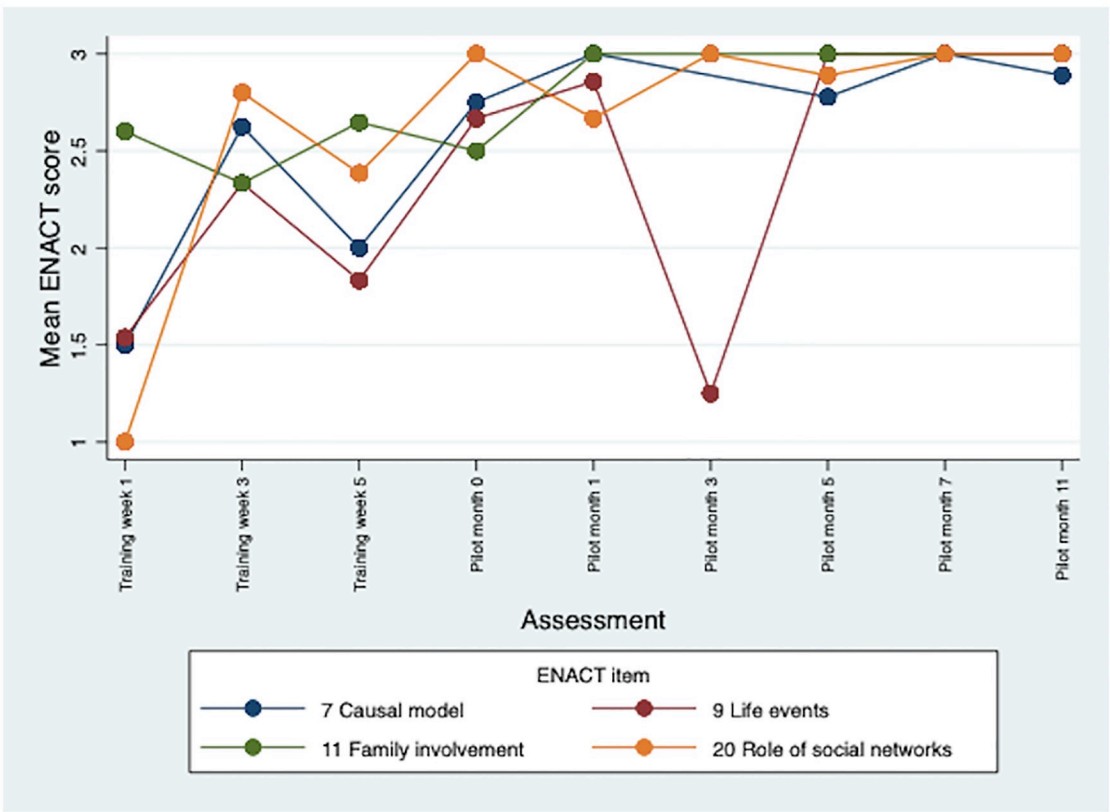

**Fig 3. Mean ENACT scores over time for external factors items.**

**Developing competence through training.** Generally, CBR workers felt they had gained much new and relevant knowledge through training, such as about the causes and treatments for schizophrenia, which would be useful for their work.

"*Now I understood what schizophrenia is and how could it occur in patients. Previously I was thinking it could come because of stress, some evil spirit or witchcraft. However now [I understand] anyone could be vulnerable to this illness as a result of any life problems.*"

(CBR worker, FGD 1, Post-training)

Several CBR workers emphasised the communication skills they had learnt during training, which for many included new techniques. This finding reflects the early improvements in 'therapeutic alliance' competencies, such as non-verbal communication, demonstrated in the ENACT assessments.

"*What makes me happy with the training was that I have been trained..to listen and respect others' views. This was an interesting thing from the training. I was working in microfinance [before]. When I was working there, refuting and attacking someone's idea was the principle. However, I have learned here to listen to the ideas of others then finally to respond for those questions. I have learned a lot from this idea. It was this experience which was the best one.*"

(CBR worker, FGD 2, Post-training)

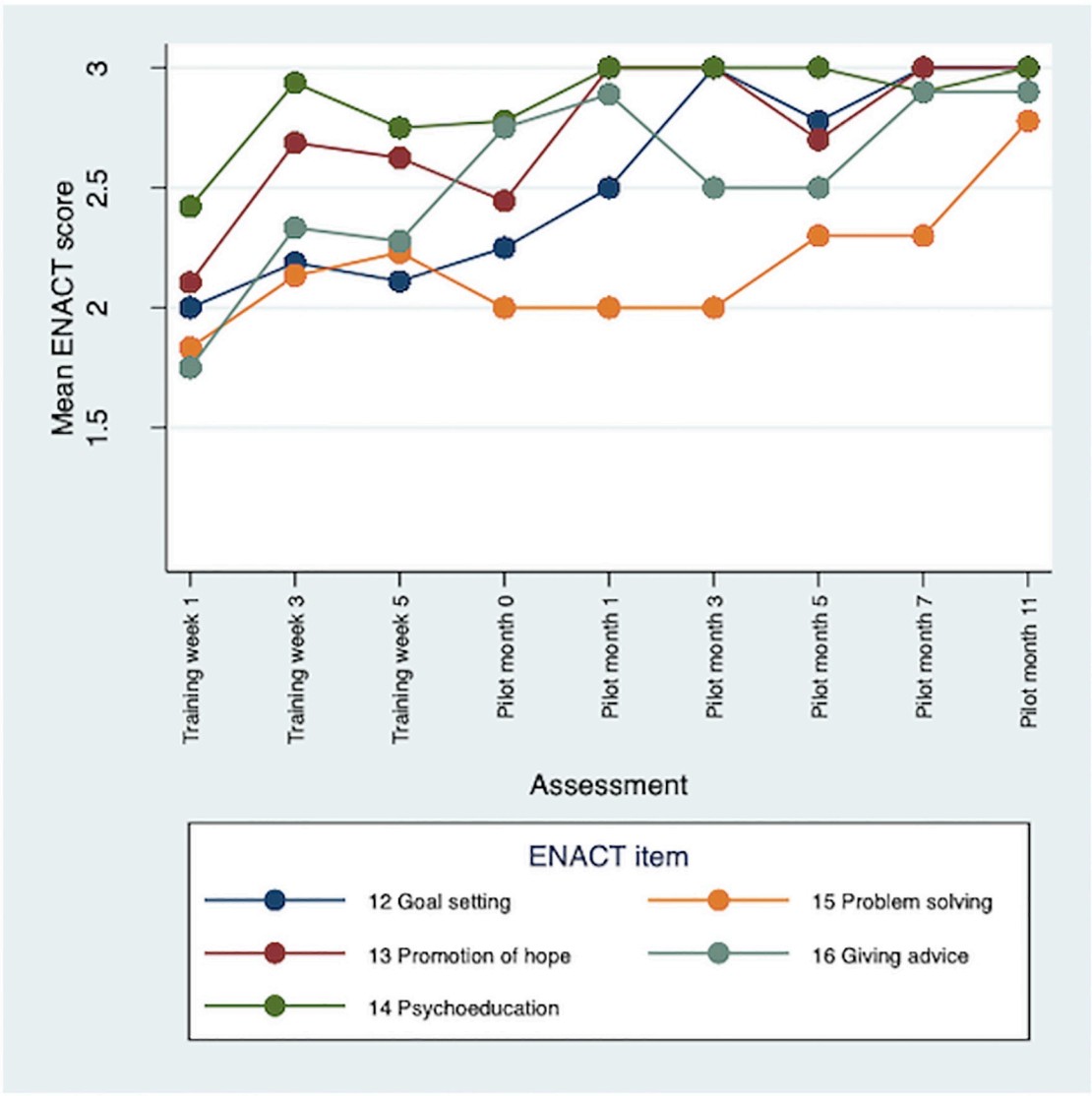

**Fig 4. Mean ENACT scores over time for other domain items.**

Whilst no teaching methods were considered redundant, role-plays had particular support from CBR workers. Getting immediate feedback was highly valued as a way to identify problem areas and ultimately improve skills.

"*The assessment which I was taking clearly indicated at which level that I am. At the next step it could indicate which things I had improved and which I had not. It could reveal at which point I didn't improved. This is very important*"

(CBR worker, FGD 1, Post-training)

CBR workers appreciated trainers who utilised the most participatory teaching styles and some noted the lack of jargon and use of Amharic language throughout.

"*Though I took other trainings before, this was special as it was easy and clear to understand because it was in Amharic and without any technical jargon in it.*"

(CBR worker, FGD 2, Post-training)

There was a unanimous opinion amongst CBR workers that the community-based practice element of training was highly valuable. The benefits of the mental health-focused components included increased understanding of the possibility of recovery and the role of medication. For some CBR workers meeting people with schizophrenia first-hand reduced concerns about working with this group:

"*The other thing is the confusion I had on how to work with people with schizophrenia. I was stressed [about] how to do the work. This stress was minimized after I went to Butajira* [regional psychiatric facility] *for the field practice. After I met the patients there, I started to calm down.*"

(CBR worker, FGD 2, Post-training)

Some CBR workers felt the generic CBR practice-focused elements were less relevant as it did not relate to mental illness. However most saw benefits, in particular how to treat families with empathy and respect, and the importance of creating strong ties with the community; for some these visits were also motivating and inspiring.

"*We have seen the passion and commitment the CBR field workers had. I also learned that I have to work in that way when I do the CBR. They helped me to know how to approach the community and the patient.*"

(CBR worker, FGD 2, post-training)

There were mixed views regarding the length of training: roughly half of CBR workers thought the training was long enough, whilst half felt it was too short. For the latter group, this meant that whilst the level of difficulty was felt to be appropriate, some topics were too not covered thoroughly.

"*The training was not too easy and also I can't also say it is too difficult. The training was appropriate for us. . . . What makes it somehow difficult was . . . the shortage of time.*"

(CBR worker, FGD 2, Post-training)

**Challenges in translating theory to practice.** One CBR worker, with previous community work experience, felt confident to deliver CBR following training. Several other CBR workers reported that they initially had little confidence in their ability to implement the theory into practice. It was only after working for some time that they gained a deeper understanding of what CBR involved and how to do the work. Facing challenges, such as relapse or family conflict, were seen as important ways for CBR workers to learn.

"*I took the five weeks training, which was believed to be enough for this work. I was trying to deliver the service using the training. However I don't feel that training is enough as it is*

*difficult to internalize the CBR properly. It is only when you do the CBR repeatedly that you could know about it.*"

(CBR worker 1, FGD 5, pilot endline)

This finding is reflected in the substantial improvements across a range of ENACT items, such as assessing functioning and goal setting, during the course of the pilot study (Table 2). One CBR worker noted the benefits of starting with the minimal workload of one family per CBR worker, before the planned progression to eight families in the RISE trial. The manual was described as comprehensive, clear and a useful learning aid. Both supervisors, but only one CBR worker, explicitly referred to consulting the manual to check facts or procedures during CBR work. The supervisors felt the flow charts were valuable in guiding actions of the CBR worker, particularly in complex situations.

"*[the CBR workers] have difficulty solving complicated problems. Hence, using the flowchart has helped us make them follow the instructions; it was helping us a lot. Whenever we encounter a situation..we always tell the CBR workers to check the flowchart.. it has been really helpful.*"

(Supervisor 2, Pilot endline)

In parallel to the increase in confidence over time, an important theme was the change in attitudes towards people with schizophrenia. Most CBR workers described how fears of violence had persisted despite the training, and several assumed their pilot participant would be aggressive. These CBR workers described how their fears were alleviated either immediately on meeting their participants, or after a few visits.

"*Even after I got the training, I was not thinking that [people with mental illness] don't attack anyone, until I begin working with them. . . . Even I had a plan to have something in my bag to defend myself. I always thought that I should defend myself [so that] I shouldn't die. However when I went there and worked with them, I don't need anything other than paper and pen.*"

(CBR worker 2, FGD 5, Pilot endline)

CBR workers extrapolated their positive experiences with the pilot participants to people with schizophrenia in general. Alongside this, CBR workers reported an increased understanding that people with schizophrenia can recover with treatment. Two CBR workers noted that whilst not all people with schizophrenia are violent, there are cases where this is true; for one of the supervisors this had implications for the safety of CBR workers. The need for an extended time period working directly with people with schizophrenia before a change in attitudes was achieved may be reflected in the relatively slower progress on 'Promotion of hope' competencies observed in the ENACT assessments.

CBR workers seemed to occupy dual roles of expert and close friend, succinctly captured by one caregiver: *"[the CBR worker] advises and explains to me like a doctor, like a brother"* (Caregiver linked to CBR worker 2; pilot month 2). Several CBR workers and participants emphasised their close relationships. This finding converges with the particularly high scores recorded for the ENACT item 'Building trust' (Table 2). For some CBR workers building this trusting relationship was central to engagement in CBR and their ability to effect change within a family.

*"Throughout our work we are communicating as good friends. . . .eh. . . That aggressiveness, that eh. . . hostility eh. . . loneliness has changed with friendship. When this has been improved, all things are changing."*

(CBR worker 1, FGD 5, pilot endline)

Equally, many participants seemed to consider CBR workers as experts, describing them as knowledgeable, able to explain clearly and give constructive advice.

*"The CBR worker has both the ability to understand and explain issues. . . .okay. . . he both explains in a way which is understandable to us and listens and understands what we tell to him. . . .eh. . . we even ask him what is not clear to us. . . .eh. . . he explains why that was in that way."*

(Person with schizophrenia linked to CBR worker 6, pilot month 2)

Most CBR participants described the calm and pleasant manner of the CBR workers, as well appreciating their ability to listen. Many participants also described how the CBR workers were motivating and had worked with them to solve their problems, for example in relation to family relationships or finding work.

*"He told us that we can count on him and call him when we experience some problems. He told us that he will try to solve the problem by cooperating with us."*

(Caregiver linked to CBR worker 6, pilot endline)

However, CBR workers sometimes met challenges in applying CBR theory into practice. Two participants described problems with the way they were given advice, one of whom complained that the CBR worker had spoken to him as if he was a child. Some participants felt they were nagged to take medication and that their concerns about side-effects were not addressed. These findings tally with the persistently lower scores on the ENACT items 'Giving advice & eliciting feedback' and 'Problem solving'. The 'Giving advice & eliciting feedback' item, in particular, focuses on collaboratively defining solutions and not lecturing clients. CBR workers who supported people with co-morbid intellectual disability criticized the lack of training on the different treatment and communication issues for this group. In addition, two CBR workers felt that had not been taught enough detail to satisfy the needs of better-educated participants.

*"The clients are not only those who are illiterate; some of the caregivers who are educated want to discuss the medications in detail. . . I feel what we gave to them was not enough. . .It was not a training which could enable us to explain about the treatment."*

(CBR worker 1, FGD 5, pilot endline)

**Sustaining competence through supervision and wellbeing.** Two linked factors emerged as important in sustaining the competence of CBR workers: supervision and wellbeing. The supervisors and some CBR workers felt that supervision had an important role in guiding CBR workers; three CBR workers said supervisors were responsive and available. CBR workers participated in a mean of ten one-to-one supervision sessions and eight group supervision sessions during the 12-month pilot study. Most CBR workers and supervisors described a

convivial relationship, more similar to peers than exhibiting a clear hierarchy (*"The relationship between the CBR workers and me is not like a worker and a boss. Rather it is a brotherly sisterly relationship."* (Supervisor 1, pilot midline)). But there were indications from both sides that this did not always function well for CBR delivery. On the one hand, one supervisor felt that CBR workers did not always ask for support in a timely manner.

> "*The CBR workers were not informing us when they faced challenging situations. They were trying to solve them on their own. . ... We constantly told them to call and notify us of the challenging situations they encounter and not to wait until we meet after a week or two. We constantly reminded them that it would be difficult for us to control these kinds of situations, if we didn't act on time.*"
>
> (Supervisor 1, pilot endline)

On the other hand, one CBR worker indicated her supervisor did not contribute much to her work and was often unavailable *("I don't get the support that I need on the time I need it"*, CBR worker 1, FGD 5, pilot endline). Another CBR worker commented that whilst supervisors were good at identifying skill gaps, refresher training was slow to materialize.

Most CBR workers reported a sense of job satisfaction and a desire to continue working in the role, despite difficult work, low pay and sometimes late salary payments.

> "*My salary is not enough but I am doing the work for my mental satisfaction.*"
>
> (CBR worker 5, FGD 5, pilot endline)

Some CBR workers and supervisors described stressful situations, often related to participants wanting to quit CBR or medication, and also sadness at the difficult circumstances of participants, or worry they would not improve. Some CBR workers had a strong sense of responsibility for their participant and were concerned they would be held accountable if improvements did not materialise.

> "*I feel distressed when I think of the unimproved health of my client. It is not only her improvement but the fear of her family's change in attitude on the CBR work. . . .eh. . . I feel stressed because of that.*"
>
> (CBR worker 3, FGD 3, pilot midline)

One CBR worker warned that without efforts to maintain personal wellbeing the role would not be sustainable.

> "*Though the situation in the community is challenging, we used to solve those problems with patience and hope. . . .eh. . . It is because if we are angry from the beginning or harming ourselves, we can't continue with that [work for] long.*"
>
> (CBR worker 3, FGD 3, pilot midline)

Strategies to maintain personal wellbeing included prayer and spending time with family members. Furthermore, peer support, which took place in group supervision sessions, but also during informal meetings, was clearly appreciated by several CBR workers. Talking with other CBR workers was a chance to gain new perspectives on how to overcome problems, or simply a relief to discuss how the work was affecting them.

"*The group work should continue with great momentum, because the eleven workers have different views on solving problems. We can get better ideas from our discussions*"

(CBR worker 8, FGD 6, pilot endline)

## Discussion

CBR workers appeared able to occupy the dual roles of expert and friend for people with schizophrenia and their families. The CBR workers' ability to strike this balance may be due to their having received targeted practical training, allowing them to deliver an intervention with tangible benefits, whilst having a physical, socio-economic and cultural proximity to the recipients, as well as training on rapport building and communication skills that avoids medical jargon. The combination of 'expert' and 'friend' suggests that lay health workers do not simply 'fill a gap' in services but may offer something unique in terms of the social connection during care of people with schizophrenia.

Whilst those with experience in community health work had an advantage, all CBR workers were able to achieve a good level of competence. CBR workers felt the mixture of training modalities worked well; role plays- including used structured competence assessment and feedback- and field work were particularly important. Whilst the training programme was a similar duration to the COPSI trial in India, one of the few comparable attempts to train lay workers in psychosocial support for people with psychosis in LMIC [7], some participants felt that five weeks was too short. Trainers had practical experience in psychiatry or CBR, matching recommendations for the training of non-specialist workers [31].

Until recently, most evaluations of the competence of lay health workers in delivering psychosocial interventions relied on self-reported measures [12,32]. More recent research employing objective competence assessments has focused on care for people with common mental disorders [17,33,34] and health professionals [15]. Our study provides novel insights into levels of competence achievable for lay health workers working with people with schizophrenia. ENACT scores varied by type of competency but, in common with studies with studies from India [17], Pakistan [34], and Uganda, Liberia and Nepal [15], good levels of competence were achieved overall. Most skills relating to therapeutic alliance, such as non-verbal communication and building trust, were developed relatively easily with training. This finding is similar to Kohrt et al.'s evaluation of primary care workers in Uganda, Liberia and Nepal [15]. Some aspects of CBR sessions, such as functioning and physical health assessment, needed enhanced guidance for assessing and recording participant responses in order for skills to improve. In contrast, Kohrt et al. found that risk assessment, explaining confidentiality and physical health assessment remained amongst the lowest scoring items after 6-to 12-months of mental health service delivery, despite targeted training and ongoing supervision [15].

Some skills, including giving advice and problem solving, were slower to develop. This finding was in common with Kohrt et al.'s evaluation of primary care workers, suggesting that difficulties did not arise in CBR workers solely due to a lack of previous healthcare training [15]. The structured problem-solving approach taught in the RISE training, and assessed using the ENACT, involved the following steps: (1) formulate and prioritize primary problem, (2) brainstorm solutions, (3) explores advantages and disadvantages, and (4) formulate an action plan. Our wider evaluation of the RISE pilot found that CBR workers successfully supported participants in a variety of ways including improving family relationships and increasing income [23]. This suggests that whilst CBR workers were not necessarily following a structured problem-solving process, and therefore scoring lower on this ENACT item, they may have successfully addressed participants' problems. To employ a systematic problem-solving approach

arguably requires more advanced skills than structured assessments (for example of substance use or physical health), or the giving of standardised information, as assessed in other ENACT items. In particular exploring the advantages and disadvantages of a proposed solution requires experience (e.g., what has worked previously with others?) and the confidence to go 'off script'. Incorporating elements from another non-specialist intervention, Problem Management Plus (PM+) [35], which has an explicit focus on problem solving steps, may support greater development of this competency.

Lower scores on the item 'Explicitly asks for feedback about the usefulness of advice', as well as the disinclination to seek and respond to CBR participants' views highlighted in the qualitative data, reflects the wider issue of minimal involvement of people with psychosis in decision-making about their care in Ethiopia [28]. These findings also indicate the general default to a medical model of lecturing clients and patients seen in many settings, even in psychosocial services [36]. The RISE pilot evaluation indicated that this interaction style might result in disengagement in care [23]. There are likewise indications that the competence of non-specialists delivering a psychological intervention for perinatal depression in India can impact on patient behaviours and, in turn, patient outcomes [33].

Stigma was raised as a potential barrier to the feasibility of lay health workers working with people with mental illness in the RISE development phase [22] and in other settings [3]. The lessening of CBR workers' stigmatizing attitudes through direct contact with people with schizophrenia in the pilot, rather than through training alone, seems to demonstrate the finding that social contact can have a powerful impact on stigma [37]. In Malawi, participation in mhGAP training led to improvements in knowledge, but not attitudes, amongst primary care workers [38]. Kohrt et al. found a potential relationship between lower stigma (i.e., greater willingness to engage with persons with mental illness) and better clinical performance in primary care workers as measured with ENACT in Nepal, Liberia and Uganda [15]. A recent anti-stigma intervention for primary care workers suggests that stigma reduction may contribute to improved competence [39]. We were unable to directly explore a relationship of stigma with the ENACT scores as our information on attitudes came only from qualitative data.

Key factors emerging as important for sustained competence of CBR workers were CBR worker wellbeing and adequate supervision. Some CBR workers described feeling stressed or distressed due to their work. Geographic and socio-cultural similarities between CBR workers and participants, whilst supporting a better experience for participants, may increase the likelihood of emotional over-involvement. Peer supervision is proposed as a useful approach to reduce stress and minimise feelings of inadequacy [17], and may also help lay workers to identify and address problems maintaining appropriate boundaries. The success of peer supervision groups in our study reflects findings from India and Pakistan indicating that this modality is acceptable and feasible for this cadre of worker [17,33,34]. The use of non-specialist supervisors was a pragmatic choice in our endeavour to design a scalable intervention, recognising the extreme shortage of mental health specialists in rural Ethiopia. However, the WHO recommends that expansion of a non-specialist mental health workforce should go hand-in-hand with increased capacity for specialist training and supervision [40]. CBR supervisors had no mental health experience and were, understandably, felt by some CBR workers to lack relevant expertise.

## Strengths and limitations

Strengths of this study include the measurement of competence at nine time points and the triangulation of qualitative and quantitative data. The ENACT instrument was originally designed to assess mental health encounters led by non-specialists in LMIC settings, and

underwent further adaptation for the local Ethiopian setting. Furthermore, the specific adaptations for CBR addressed the criticism that tools to assess psychological therapy quality are often too generic to capture the disorder and treatment specific strategies which underpin their mode of action [8]. We also successfully demonstrated that non-specialist raters (e.g., lay supervisors) are able to detect changes in competence over time and between CBR workers, which is an asset in resource-constrained settings.

Despite these advantages, because of the focus on the one-on-one helper-client interaction, the ENACT did not capture other CBR competencies, such as delivering community mobilisation and promotion of participation in economic activities. Ability to instigate community linkages is recognised as a component of competency in mental healthcare in sub-Saharan Africa [41] and disability-inclusive development [42] more broadly. However, there are few examples of how such competencies may be assessed in practice. Process data collected in the RISE pilot and trial assessed whether mobilisation tasks were completed, but not how well [23]. Due to the small numbers of participants, inter-rater reliability was not calculated, only a descriptive analysis of scores was presented and we did not link competency to participant outcomes. CBR workers were known to the ENACT assessors, who were CBR supervisors, and this may have biased the ratings. Due to logistical constraints, structured role play assessments were not conducted during the pilot period meaning that ENACT-E ratings were not truly comparable over the training and pilot periods. However, quality ratings of actual CBR sessions are arguably the more pertinent assessment modality once CBR workers have started in the role, as they reveal how skills are applied in practice.

## Implications and recommendations

Our findings are largely supportive of the use of lay health workers for the delivery of CBR for people with schizophrenia. Overall, good levels of competence were achieved, and moreover, the lay person status of CBR workers may have made a unique contribution to the quality of the intervention. Multimodal training, including role plays, community-based visits and group discussions, was strongly endorsed by CBR workers. However, we recommend that lay health workers also need a period of on the job training or apprenticeship, ideally over several months, in order to acquire more advanced therapeutic skills for the care of people with psychosis. These skills may be of critical importance in ensuring engagement in both CBR and facility-based care, and therefore improved patient outcomes. We also propose a highly structured intervention with clear protocols and flowcharts is adopted to ensure good quality of CBR delivery. Future interventions should be alert to the possibility of distress or stress amongst lay health workers working with people with schizophrenia. Peer supervision appears to be a powerful means to mitigate these experiences. Rigorous risk assessments and procedures to protect lay worker safety are also paramount. Our use of non-specialist supervision had limitations. Specialist supervisors, for example psychiatric nurses or mental health social workers, should therefore be considered in any future implementation of CBR for schizophrenia in LMIC. Other mechanisms to address logistical constraints and shortage of specialist supervisors could be explored. In Pakistan, Rahman et al. found no significant difference in ENACT scores between community health workers supervised from a distance using a training application, and those supervised face-to-face [16].

We used the ENACT to measure and improve CBR worker competence in two key ways during the RISE CBR worker training and pilot and we recommend that future feasibility studies adopt a similar approach. First, to identify weaker areas across the cohort of CBR workers. These findings were used to target top-up training to areas of need and iteratively amend CBR documentation and protocols. Second, to provide immediate structured feedback to CBR

workers based on observed role-plays and clinical encounters. The ENACT can also be used as a treatment fidelity measure [13]. Exploratory analysis of RISE trial outcomes will consider the relationship between CBR worker competence and patient functioning. Since conducting this study, the WHO has begun building the Ensuring Quality in Psychological Support (EQUIP) platform. This platform will support use of a suite of competency assessments, including the ENACT, for programmatic and research settings globally [19]. A promising avenue, drawing on the success of peer supervision in the RISE pilot, may be the use of the peer-rated ENACT during training and intervention delivery. This approach has been used successfully alongside non-specialist provider delivered psychological interventions for perinatal depression, harmful drinking and severe depression in India using the Therapy Quality Scale or other bespoke instruments [17,33,43,44]. However, expert support may be needed to ensure feedback is structured and reliable [33]. Observation of some home visits by a supporting CBR worker could also be a relatively resource-light means of ensuring broader intervention fidelity, that would have the additional benefit of minimising any safety concerns associated with lone visits.

## Conclusion

Our quantitative and qualitative findings indicate that it is possible to train lay people with little or no prior experience to deliver a psychosocial intervention for people with schizophrenia in a low-income country setting. Some aspects of a high quality CBR intervention, such as problem-solving skills, only developed after a prolonged period of work experience. Repeated ENACT assessments were successfully used to identify areas requiring improvement and guide on-going training for CBR workers.

## Supporting information

**S1 Appendix. RISE training outline.**
(DOCX)

**S2 Appendix. RISE CBR worker training manual (English version).**
(PDF)

**S3 Appendix. ENACT-E.**
(PDF)

## Acknowledgments

We would like to acknowledge the valuable contributions made by all CBR workers and CBR supervisors.

## Author Contributions

**Conceptualization:** Laura Asher, Brandon A. Kohrt, Charlotte Hanlon.

**Data curation:** Laura Asher, Rahel Birhane.

**Formal analysis:** Laura Asher.

**Investigation:** Laura Asher, Barkot Milkias, Benyam Worku, Alehegn Habtamu.

**Methodology:** Laura Asher, Solomon Teferra, Brandon A. Kohrt, Charlotte Hanlon.

**Project administration:** Laura Asher, Rahel Birhane, Alehegn Habtamu.

**Resources:** Brandon A. Kohrt.

**Supervision:** Laura Asher, Rahel Birhane.

**Writing – original draft:** Laura Asher.

**Writing – review & editing:** Laura Asher, Rahel Birhane, Solomon Teferra, Barkot Milkias, Benyam Worku, Alehegn Habtamu, Brandon A. Kohrt, Charlotte Hanlon.

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
