## [Decision Letter · Decision Letter 0]

18 Nov 2020

PONE-D-20-30256

“Like a doctor, like a brother”: achieving competence amongst lay health workers delivering community-based rehabilitation for people with schizophrenia in Ethiopia

PLOS ONE

Dear Dr. Asher,

Thank you for submitting your manuscript to PLOS ONE. After careful consideration, we feel that it has merit but does not fully meet PLOS ONE’s publication criteria as it currently stands. Therefore, we invite you to submit a revised version of the manuscript that addresses the points raised during the review process.

We look forward to receiving your revised manuscript.

Kind regards,

Danuta Wasserman

Academic Editor

PLOS ONE

Journal Requirements:

2. Please provide additional information about your teaching intervention, such as detailed curriculum, description of texts or methods used, or supporting educational material that would allow others to replicate your study. If materials, methods, and protocols are well established, authors may cite articles where those protocols are described in detail, but the submission should include sufficient information to be understood independent of these references (https://journals.plos.org/plosone/s/submission-guidelines#loc-materials-and-methods).

Reviewers' comments:

Reviewer's Responses to Questions

**Comments to the Author**

1. Is the manuscript technically sound, and do the data support the conclusions?

Reviewer #1: Yes

2. Has the statistical analysis been performed appropriately and rigorously? 

Reviewer #1: N/A

3. Have the authors made all data underlying the findings in their manuscript fully available?

Reviewer #1: Yes

4. Is the manuscript presented in an intelligible fashion and written in standard English?

Reviewer #1: Yes

5. Review Comments to the Author

Reviewer #1: Overall, this is a very interesting paper, documenting how lay people were trained and supervised in order to work with people with schizophrenia. It is documented by scorings of competences and focus group interviews. The paper is highly relevant as it documents how lay people can be trained and learn to help mental health services, which are sparse.

The introduction is interesting, it is clearly written and includes relevant citations and considerations.

The study is highly relevant as it addresses unmet needs in the population in LMIC countries and it demonstrates an elaborated model for training lay people. The authors conducted a mixed method study of lay people trained in mental health treatment. The CBR worker competence was measured using the ENhancing Assessment of Common Therapeutic factors (ENACT) rating scale.

It would be interesting to know which proportion of the patients were on medication?

The legends of table 1 should mention the sample and explain ENACT, in order to make it possible to read the table without consulting the text in details.

I am not an expert in qualitative research, but I do know that nVIVO is often used and is considered a gold standard instrument. If necessary due to space limitations, some of the cited phrases could be left out

The discussion adequately discusses the results and future interventions

The main challenge will be to make sure that lay people do not practice work that is difficult or dangerous for themselves and for the patients. This could involve emotional overinvolvement, taking over too much responsibility, and ending up in a role like a relative. Lay people will be faced with unmet needs of the patients, and how to decide what to do? Procedures must make sure that the lay people can continue their own personal life outside working hours without too much worry and concern regarding their job. Also, idiosyncratic practice where a lay person starts to work mainly guided by own opinion or therapeutic ideas should be addressed. It could also be considered to work in pairs in order to make sure that practice is visible?

6. PLOS authors have the option to publish the peer review history of their article (what does this mean?). If published, this will include your full peer review and any attached files.

Reviewer #1: **Yes: **merete nordentoftno COI

---

## [Author Response · Author response to Decision Letter 0]

9 Dec 2020

Thank you for the opportunity to revise the above manuscript. Please find the responses below (note that line/page references refer to the tracked changes version of the manuscript).

Journal requirements 

Comment: Ensure that your manuscript meets PLOS ONE's style requirements, including those for file naming

Response: Amendments have been made to meet the guidelines

Comment: Please provide additional information about your teaching intervention, such as detailed curriculum, description of texts or methods used, or supporting educational material that would allow others to replicate your study.

Response: There is a summary of the RISE training programme in the methods lines 185-211, which we feel is appropriate detail for the body of the paper. I had provided a link to the 200 page RISE training manual, but I have now added this as an appendix instead. There is an overview of the manual contents in the main text.

S1 Appendix provides an overview of the RISE training programme. To this file I have added the following text signposting to the detailed training materials ‘Further training resources, including powerpoint slides and training activities in English and Amharic, are available on request laura.asher@nottingham.ac.uk’. The training materials consist of 46 powerpoint presentations, each in English and Amharic, 32 handouts/exercises, plus videos and other materials, so it is not practical to link these directly to the paper.

Comment: Amend your list of authors on the manuscript to ensure that each author is linked to an affiliation. 

Response: Minor amendments made to author affiliations. Each author has at least one affiliation.

Comment: Include captions for your Supporting Information files at the end of your manuscript

Response: These have been added

Reviewer

Comment: It would be interesting to know which proportion of the patients were on medication?

Response: We have added details on the pilot participants including medication use, “At recruitment into the pilot, half of participants were treatment naïve and only one participant was taking anti-psychotic medication. Several participants went on to take anti-psychotic medication for varying periods during the 12 month pilot study”. See lines 251-261.

Comment: The legends of table 1 should mention the sample and explain ENACT, in order to make it possible to read the table without consulting the text in details.

Response: We have amended the titles and footnotes of Tables 1 and 2.

Comment: If necessary due to space limitations, some of the cited phrases could be left out

Response: We feel all the included quotes would ideally remain in the paper to adequately represent the rich qualitative data supporting our analysis. However if the editor feels the overall length needs reducing, we can omit some of these.

Comment: The main challenge will be to make sure that lay people do not practice work that is difficult or dangerous for themselves and for the patients. This could involve emotional overinvolvement, taking over too much responsibility, and ending up in a role like a relative. Lay people will be faced with unmet needs of the patients, and how to decide what to do? Procedures must make sure that the lay people can continue their own personal life outside working hours without too much worry and concern regarding their job.

Response: We agree these are important challenges. We have added text to the methods to describe how CBR wellbeing and safety was assured (lines 238-240 and 269-273) : “The training covered approaches to support CBR wellbeing, including keeping appropriate boundaries with participants, and how safety would be assessed and maintained.” And “All home visits were made with caregivers present and CBR workers were required to carry a mobile phone at all times. CBR supervisors made regular risk assessments of the home visit environment. Additional safety procedures, such as only permitting joint visits with supervisors, were implemented according to the assessment outcome.”

We have also added reflections on these issues in the discussion, lines 845-850: “Geographic and socio-cultural similarities between CBR workers and participants, whilst supporting a better experience for participants, may increase the likelihood of emotional over-involvement. Peer supervision is proposed as a useful approach to reduce stress and minimise feelings of inadequacy (17), and may also help lay workers to identify and address problems maintaining appropriate boundaries.” And lines 910-911 “Rigorous risk assessments and procedures to protect lay worker safety are also paramount.” 

Comment: Also, idiosyncratic practice where a lay person starts to work mainly guided by own opinion or therapeutic ideas should be addressed. It could also be considered to work in pairs in order to make sure that practice is visible?

Response: We agree this is another important challenge and interesting potential solution- we have added reflections on this in the discussion lines 939-942 “Observation of some home visits by a supporting CBR worker could also be a relatively resource-light means of ensuring broader intervention fidelity, that would have the additional benefit of minimising any safety concerns associated with lone visits.” 

In addition CBR supervisors did conduct unannounced observed home visits every two months (when ENACT assessements were completed), which is mentioned in the methods (lines 266-267).

---

## [Editor Report · Decision Letter 1]

15 Jan 2021

“Like a doctor, like a brother”: achieving competence amongst lay health workers delivering community-based rehabilitation for people with schizophrenia in Ethiopia

PONE-D-20-30256R1

Dear Dr. Asher,

We’re pleased to inform you that your manuscript has been judged scientifically suitable for publication and will be formally accepted for publication once it meets all outstanding technical requirements.

Kind regards,

Danuta Wasserman

Academic Editor

PLOS ONE

Additional Editor Comments (optional): Important topic. The authors addressed relevantly the methods used and the data analysis. 
---

## [Editor Report · Acceptance letter]

12 Feb 2021

PONE-D-20-30256R1 

“Like a doctor, like a brother”: achieving competence amongst lay health workers delivering community-based rehabilitation for people with schizophrenia in Ethiopia 

Dear Dr. Asher:

I'm pleased to inform you that your manuscript has been deemed suitable for publication in PLOS ONE. Congratulations! Your manuscript is now with our production department. 

Kind regards, 

on behalf of

Dr. Danuta Wasserman 

Academic Editor

PLOS ONE